Body mass index influences the plasma glucose concentration during iatrogenic hypoglycemia in people with type 2 diabetes mellitus: a cross-sectional study

Cheng Po Chung 1
Hsu Shang Ren 1
Tu Shih Te 10836@cch.org.tw luukskyw@yahoo.com 1
Cheng Yun Chung 2
Liu Yu Hsiu 3
1 Division of Endocrinology and Metabolism, Department of Internal Medicine, Changhua Christian Hospital , Changhua City , Taiwan
2 Department of Radiology, Taichung Veterans General Hospital , Taichung , Taiwan
3 Department of Accounting and Information Systems, National Taichung University of Science and Technology , Taichung , Taiwan
Foti Daniela
Electronic publication date: 2018 Feb 15
Publication date: 2018
Volume: 6
Electronic Location ID: e4348
Received 2017 Dec 15; Accepted 2018 Jan 19
Copyright: ©2018 Cheng et al.
Copyright year: 2018
Copyright holder: Cheng et al.
License: This is an open access article distributed under the terms of the Creative Commons Attribution License, which permits unrestricted use, distribution, reproduction and adaptation in any medium and for any purpose provided that it is properly attributed. For attribution, the original author(s), title, publication source (PeerJ) and either DOI or URL of the article must be cited.
License URL: https://creativecommons.org/licenses/by/4.0/

Keywords: Type 2 diabetes mellitus, Body mass index, Complications, Hypoglycemia

Funding: The authors received no funding for this work.

==============================
Background

Hypoglycemia occurs in an appreciable number of individuals with type 2 diabetes mellitus (T2DM) who are receiving glycemic therapy. Iatrogenic hypoglycemia induces not only complications but also a substantial medical expense. Intervention for relevant risk factors may help avert severe hypoglycemia and enhance quality of life in at-risk individuals. This study investigates the relationship between body mass index (BMI) and plasma glucose concentration during iatrogenic hypoglycemia in people with T2DM.

Methods

Enrollment criteria were people above 20 years of age, with existing diagnosis of T2DM, a documented plasma glucose level ≤70 mg/dL, and acute cognitive impairment requiring hospitalization. Participants were classified into two groups according to their BMI. Specifically, lower BMI subgroup denotes individuals whose BMI fall within lower half of the study population, and vice versa. Plasma glucose concentration, length of hospital stay, and serum electrolyte level at hospitalization were compared between these BMI subgroups. Moreover, multivariate regression analysis was performed to identify covariates associated with plasma glucose level during iatrogenic hypoglycemia.

Results

This study enrolled 107 participants for whom 54 were assigned to a higher BMI subgroup and the remainder to a lower BMI subgroup. People with lower BMI harbored substantially reduced plasma glucose concentration during iatrogenic hypoglycemia compared to those with higher BMI (30.1 ± 9.6 mg/dL vs. 38.4 ± 12.3 mg/dL, P < 0.001). Nonetheless, the length of stay (6.2 ± 4.6 days vs. 5.7 ± 4.0 days, P = 0.77) and serum potassium level (3.7 ± 0.9 meq/L vs. 3.9 ± 0.8 meq/L, P = 0.14) were comparable between subgroups. Multivariate regression analysis identified BMI as a determinant of plasma glucose concentration in diabetic individuals with iatrogenic hypoglycemia (β coefficient: 0.72, P = 0.008).

Discussion

In individuals with T2DM who experience severe iatrogenic hypoglycemia, BMI influences the plasma glucose level at hospitalization. People with lower BMI harbored appreciably reduced plasma glucose concentration relative to their higher BMI counterparts. In lower weight people, therefore, appropriate dosing of antidiabetic medications, frequent self-monitoring of blood glucose level and adequate nutritional support may help avert more severe hypoglycemia. Overall, BMI potentially influences the severity of iatrogenic hypoglycemia in people with T2DM.

Introduction

Hypoglycemia occurs in an appreciable number of individuals with type 2 diabetes mellitus (T2DM) who are receiving antidiabetic therapy (Shafiee et al., 2012). Iatrogenic hypoglycemia induces not only complications but also a substantial medical expense related to hospitalization (Rhee et al., 2016). Prevention of hypoglycemia in diabetes is therefore an integral component of treatment (Clayton, Woo & Yale, 2013). Importantly, intervention for relevant risk factors may reduce severe hypoglycemia and enhance quality of life in people with T2DM.

Iatrogenic hypoglycemia is associated with clinical complications. Elderly people with recurrent hypoglycemia are at risk of cognitive impairment (McNay & Cotero, 2010). Severe hypoglycemia also contributes to a higher incidence of cardiovascular event and mortality (Hanefeld, Frier & Pistrosch, 2016). Moreover, risk of traumatic injury is notably increased in patients who experience severe hypoglycemia (Kachroo et al., 2015).

Epidemiologic studies have identified several risk factors for hypoglycemia in people with T2DM. Stringent glycemic targets are closely linked to the incidence of hypoglycemia (Lipska et al., 2013). In addition, severe hypoglycemia occurs more frequently in the elderly, presumably due to lack of symptom recognition (Abdelhafiz et al., 2015). Studies have also implicated diabetes duration as a risk factor, as demonstrated by progressive deterioration of hypoglycemic counterregulatory mechanism in people with longstanding disease (Dailey et al., 2013; Amiel et al., 2008).

The body mass index (BMI), as defined by dividing the weight in kilograms by the square of the height in meters, may be an important but overlooked risk factor for severe hypoglycemia. People with T2DM who have a lower BMI are also likely to have lower hepatic glycogen stores and this can diminish the secretion of glucose counterregulatory hormones during hypoglycemia (Winnick et al., 2016). Reduced counterregulatory hormones may compromise an individual’s capacity to stabilize blood glucose level during fasting (Izumida et al., 2013). Furthermore, lower BMI in elderly individuals may reflect underlying frailty (Lee et al., 2014), which impairs both the recognition of and behavioral defense against hypoglycemia.

This study investigates the relationship between BMI and plasma glucose concentration during iatrogenic hypoglycemia in people with T2DM.

Materials and Methods

This is a cross-sectional study conducted at Changhua Christian Hospital in central Taiwan. Individuals hospitalized at the Endocrinology ward from September 2011 to August 2017 were assessed for eligibility. Enrollment criteria were people above 20 years of age, with existing diagnosis of T2DM, a documented plasma glucose level ≤70 mg/dL (Cryer, 2015), and acute cognitive impairment that required hospitalization.

Candidates were excluded if they had chronic infection, traumatic injury, acute cerebro- or cardiovascular event, organ failure, or whose hypoglycemia was considered independent of antidiabetic medications. Moreover, people with malignancy, eating disorders, and previous gastrointestinal surgery were ineligible. The study was approved by the Institutional Review Board of Changhua Christian Hospital (CCH IRB number: 171105). Written consent to participate in the study was provided by the patients’ next of kin.

Participants were assigned to two equally sized subgroups according to their BMI. Specifically, lower BMI subgroup denotes individuals whose BMI fall within lower half of the study population, and vice versa. Blood tests were performed at hospitalization except for glycosylated hemoglobin A1c (HbA1c), which was extracted from existing laboratory data prior to the hypoglycemic event. Serum HbA1c was measured by ion-exchange high-performance liquid chromatography using BioRad VARIANT™ II Turbo system. Serum biochemistry including glucose, creatinine, and potassium were measured by Beckman Coulter UniCel DxC 800 Synchron™ Clinical Systems (Beckman Coulter, Brea, CA, USA). The analytical precision for serum glucose is within 2 mg/dL. Prescription details were collected from electronic medical records. The enrollment process is illustrated in Fig. 1.

Figure 1 Enrollment process of the study.

Demographic data between BMI subgroups were compared using Mann–Whitney U test for continuous variables and Pearson’s χ2 test for categorical variables. The plasma glucose concentration, serum potassium, and length of hospital stay were compared using Mann–Whitney U test. Multivariate regression analysis was performed to identify predictors of plasma glucose level at hospitalization. A two-tailed P value of less than 0.05 indicated statistical significance. Analysis was performed using IBM SPSS version 22.0 (IBM SPSS Statistics for Windows, Armonk, NY, USA).

Results

The study enrolled 107 participants with T2DM who were hospitalized due to severe iatrogenic hypoglycemia. As shown in Table 1, demographic characteristics including age (76.5 ± 13.8 years vs. 73.8 ± 11.7 years, P = 0.172), preceding HbA1c (6.7 ± 1.1% vs. 6.8 ± 1.3%, P = 0.52), duration of diabetes (7.6 ± 4.5 years vs. 9.3 ± 4.9 years, P = 0.051), and serum creatinine (1.6 ± 1.0 mg/dL vs. 1.9 ± 1.6 mg/dL, P = 0.336) were comparable between BMI subgroups. Moreover, the proportion of participants using antidiabetic medication commonly implicated in iatrogenic hypoglycemia, such as insulin (26% vs. 32%, P = 0.31) or sulfonylurea (65% vs. 64%, P = 0.55), were also similar.

Table 1 Demographic characteristics of the body mass index subgroups.

Variables	Lower BMI (n = 54)	Higher BMI (n = 53)	P value	
Age (years)	76.5 ± 13.8	73.8 ± 11.7	0.172	
Sex (Female)	29 (54%)	29 (55%)	0.53	
BMI (kg/m2)	20.1 ± 1.78	27.0 ± 3.90	<0.001	
HbA1c (%)	6.7 ± 1.1	6.8 ± 1.3	0.52	
Creatinine (mg/dL)	1.60 ± 1.00	1.91 ± 1.63	0.336	
Duration of diabetes (years)	7.6 ± 4.5	9.3 ± 4.9	0.051	
Use of insulin	14 (26%)	17 (32%)	0.31	
Use of sulfonylurea	35 (65%)	34 (64%)	0.55	
Notes.

Data are expressed as mean with standard deviation for continuous variables and number (%) for categorical variables. Continuous variables were compared using the Mann–Whitney U test for independent samples.

BMI body mass index

HbA1c glycated hemoglobin A1c

As demonstrated in Table 2, participants with lower BMI harbored substantially reduced plasma glucose concentration compared to those with higher BMI during iatrogenic hypoglycemia (30.1 ± 9.6 mg/dL vs. 38.4 ± 12.3 mg/dL, P < 0.001). Nonetheless, individuals with lower BMI did not have an appreciably longer length of stay compared to their higher weight counterparts (6.2 ± 4.6 days vs. 5.7 ± 4.0 days, P = 0.77). Furthermore, mean serum potassium levels were similar between subgroups (3.7 ± 0.9 meq/L vs. 3.9 ± 0.8 meq/L, P = 0.14).

Table 2 Clinical features of the body mass index subgroups.

Variables	Lower BMI (n = 54)	Higher BMI (n = 53)	P value	
Plasma glucose concentration (mg/dL)	30.1 ± 9.61	38.4 ± 12.3	<0.001	
Length of stay (days)	6.2 ± 4.6	5.7 ± 4.0	0.77	
Potassium (mEq/L)	3.7 ±  0.86	3.9 ± 0.80	0.14	
Notes.

Data are expressed as mean with standard deviation for continuous variables. Continuous variables were compared using the Mann–Whitney U test for independent samples.

BMI body mass index

Multivariate regression analysis identified covariates that potentially influence the plasma glucose concentration at hospitalization. The standardized coefficient of each independent variable is listed in Table 3. As can be seen, BMI was significantly related to plasma glucose concentration during iatrogenic hypoglycemia in people with T2DM (β coefficient: 0.72, P = 0.008) after adjusting for confounding variables.

Table 3 Multivariate regression analysis of covariates associated with plasma glucose concentration during iatrogenic hypoglycemia.

Covariates	β coefficient	P value	
Age (years)	0.014	0.88	
BMI (kg/m2)	0.72	0.008	
HbA1c (%)	−1.18	0.26	
Creatinine (mg/dL)	−1.1	0.21	
Duration of diabetes (years)	−0.037	0.88	
Notes.

BMI body mass index

HbA1c glycated hemoglobin A1c

Discussion

People with T2DM are vulnerable to the detrimental effect of hypoglycemia, which may become a limiting factor in antidiabetic therapy (Seaquist et al., 2013). Apart from requiring the assistance of caregivers, severe hypoglycemia also induces harmful cardiac arrhythmia and functional brain failure (Chow et al., 2014). Moreover, hypoglycemia-associated autonomic failure can impair the physiologic and behavioral defense against a subsequent hypoglycemic event (Cryer, 2013a; Cryer, 2013b).

The observation in this study that lower weight people with T2DM had reduced plasma glucose concentration during iatrogenic hypoglycemia may be attributable to attenuated glucose counterregulatory mechanisms. People with lower BMI, perhaps reflecting less availability of hepatic glycogen, may have diminished secretion of glucagon and epinephrine (Winnick et al., 2016), resulting in inadequate hepatic glucose production during iatrogenic hypoglycemia. Furthermore, the effect of glucagon may be compromised in people with inadequate glycogen since this hormone raises blood glucose level through hepatic glycogenolysis (Melmed et al., 2016).

Moreover, unintentional weight loss may reflect frailty and functional disability (Xue, 2011). In lower weight people, delayed recognition of hypoglycemia may partly explain their appreciably lower plasma glucose level at hospitalization. People with lower BMI may therefore have limited ability to counteract hypoglycemia due to reduced secretion of glucose counterregulatory hormones, as discussed previously, and functional disability that leads to hypoglycemic unawareness.

Adipose tissue modifies insulin sensitivity through the production of adipokines (Waki & Tontonoz, 2007; Fasshauer & Blüher, 2015). Weight loss improves insulin sensitivity by decreasing free fatty acid mobilization and by changing adipokine profile in obesity (Schenk et al., 2009; Greco et al., 2014). Moreover, intentional weight loss in T2DM correlates with lower fasting plasma glucose concentration (Wing et al., 2011). Weight loss is therefore an established risk factor for iatrogenic hypoglycemia in diabetes (Melmed et al., 2016). In clinical practice, BMI may indirectly mirror an individual’s insulin sensitivity and subsequent risk of hypoglycemia. Therefore, dynamic change in body weight during glycemic treatment may require a corresponding adjustment in therapeutic regimen.

Intriguingly, although participants with lower BMI harbored reduced plasma glucose level at hospitalization, length of stay and serum potassium level were similar to their higher BMI counterparts. In other words, hypoglycemia may not cause immediately perceivable complications. Nonetheless, plasma glucose level below 30 mg/dL has been linked to permanent brain injury in an animal model (Oyer, 2013), and lower weight people in this study with a mean blood glucose close to this level were at risk of long-term neurologic damage.

Several implications arise from the study’s finding that lower weight participants had reduced plasma glucose level during iatrogenic hypoglycemia. Hypoglycemia in diabetes involves a combination of therapeutic insulin excess and compromised physiologic defense (Cryer, 2013a; Cryer, 2013b). Appropriate dosing of antidiabetic medications, especially sulfonylurea and insulin (Heller et al., 2007), is prudent for people with lower BMI. Lower weight people may also benefit from less stringent treatment target, frequent self-monitoring of blood glucose (SMBG), and continuous glucose monitoring (CGM) to detect asymptomatic hypoglycemia (Cryer, 2014). Nutritional support to increase hepatic glycogen in at risk individuals may enhance physiologic defense against more severe hypoglycemia (Décombaz et al., 2011). Moreover, since the glucose-raising efficacy of glucagon is unreliable in people with inadequate glycogen, an alternative method may be necessary to restore normoglycemia in underweight patients.

The design of this study has limitations. To be hospitalized for treatment, participants obviously circumvented lethal complications such as cardiac arrhythmia, which may lead to selection bias. Furthermore, participants may have initially received management for hypoglycemia at home, and blood tests at hospitalization may not reveal the lowest plasma glucose concentration. A longer observation time may be necessary to identify potential complications associated with severe hypoglycemia. Hypoglycemic risk may also relate to the dose of antidiabetic medications, which was not addressed by the study. Moreover, neither hepatic glycogen quantity nor counterregulatory hormone level was measured, both of which may influence the severity of iatrogenic hypoglycemia.

Conclusions

Lower weight individuals with T2DM harbored reduced plasma glucose concentration during iatrogenic hypoglycemia. People with lower BMI may have compromised defense against iatrogenic hypoglycemia due to reduced secretion of counterregulatory hormones and functional disability. Appropriate dosing of antidiabetic medications, individualized treatment target, frequent SMBG and CGM technology may help avert more severe hypoglycemia in people with lower BMI. Ultimately, nutritional support to increase hepatic glycogen may defend lower weight patients against severe hypoglycemia.

Supplemental Information

Data S1 Dataset for the study population

Click here for additional data file.

Additional Information and Declarations

Competing Interests

Author Contributions

Human Ethics

Data Availability

The authors declare there are no competing interests.

Po Chung Cheng conceived and designed the experiments, performed the experiments, contributed reagents/materials/analysis tools, wrote the paper, reviewed drafts of the paper.

Shang Ren Hsu and Shih Te Tu conceived and designed the experiments, performed the experiments, wrote the paper, reviewed drafts of the paper.

Yun Chung Cheng analyzed the data, wrote the paper, prepared figures and/or tables, reviewed drafts of the paper.

Yu Hsiu Liu analyzed the data, contributed reagents/materials/analysis tools, wrote the paper, prepared figures and/or tables, reviewed drafts of the paper, qualified statistician.

The following information was supplied relating to ethical approvals (i.e., approving body and any reference numbers):

The study was approved by the Institutional Review Board of Changhua Christian Hospital (CCH IRB number: 171105).

The following information was supplied regarding data availability:

The raw data is provided as a Supplemental File.

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
