# Peer review of "Body mass index influences the plasma glucose concentration during iatrogenic hypoglycemia in people with type 2 diabetes mellitus: a cross-sectional study"

_PeerJ, doi:10.7717/peerj.4348_

## Round 0.1 · original submission · Major Revisions

Authors should address the issues about statistics raised by both reviewers, and improve the explanations of some concepts recalled by both reviewers.

Reviewer 1 ·

Basic reporting

See below

Experimental design

See below

Validity of the findings

See below

Additional comments

In this paper, the authors demonstrate that in individuals with T2DM who experience severe iatrogenic hypoglycemia, BMI influences plasma glucose levels at hospitalization. The paper is susceptible for publication, but I have some comments that need to be addressed before by the authors:

1. Since some patients are young and treated with insulin, the authors should specify how type 1 diabetes has been excluded;
2. Patients affected by any organic disease impacting on BMI must be excluded;
3. Methods for the determination of glucose, HbA1c, creatinine, and potassium should be indicated, as well as quality of the analytical performance (precision).
4. By analyzing the file data, it emerges that all continuous variables have a non-normal distribution. Thus, non parametric tests must be employed;
5. In linear regression analysis, the impact of BMI must be adjusted for other confounding factors.
6. In the discussion: line 180, “glycemic therapy” is not clear.
7. Line 204: The Authors should briefly indicate the role of fat tissue in the production of adipokines, whose role is important in insulin sensitivity and inflammation. (Waki H, Tontonoz P, Annu Rev Pathol 2007; Fasshauer M, Bluher M, Trends Pharmacol Sci, 2015).
8. Line 205 could be better expressed as follows: “Weight loss improves insulin sensitivity by decreasing free fatty acid mobilization, and by changing adipokine profile in obesity (Schenk et al., 2009; Greco M, et al, Mediators Inflamm 2014), and intentional weight loss in T2DM, correlates with lower fasting plasma glucose concentration (Wing et al., 2011)”.
9. References should be listed in alphabetical order.

Reviewer 2 ·

Basic reporting

The manuscript is well written and associates BMI with the nadir of hypoglycemia upon admittance to the hospital for hypoglycemia-care. The study is observational and not an intervention, however, great care was taken to adjust for confounding variables where applicable. The main finding of this work is that individuals with lower BMI tend to have lower plasma glucose levels (30 vs 38 mg/dL) and that this relationship is quite strong (B-coefficient of 0.72). I have a few minor suggestions that may improve the presentation of the data and the discussion of the results, however, overall I find this manuscript to be of good scientific merit.
Throughout the manuscript, the authors refer to “energy deficit and glycogen depletion.” It should be noted, however, that unlike rodents, humans do not experience glycogen “depletion.” In fact, even after days or weeks of fasting, glycogen is still present in the liver. Furthermore, Hendrick and Cherrington (1990, AJP) showed that in dogs, glycogen is present in the liver after a 7d fast and that when their livers were exposed to glucagon, a “normal” amount of glycogen was mobilized, despite the liver glycogen concentrations. Therefore, in the context of your conclusions, I am of the opinion that the proposed mechanism whereby “BMI is associated with chronic energy deficit and glycogen depletion” would cause “more severe hypoglycemia” is slightly inaccurate. On the other hand, in the paper by Winnick et al., that you cite, they observed that low liver glycogen, such as what you are suggesting is present with lower BMI, is associated with diminished secretion of the counterregulatory hormones glucagon and epinephrine. By reducing glucagon and epineiphrine secretion, it would be expected to lower hepatic glucose production and make the glucose nadir lower than what was occurs in the high-BMI group. I am of the opinion that in order to provide a speculative mechanism of action, this is the most consistent with your data.
In the discussion, the use of the phrase “tolerance of hypoglycemia….” is of interest. Perhaps the authors mean to say “ability to counterregulate hypoglycemia….” Or perhaps “tolerance to hyperinsulinemia….” Would be a better description?

Experimental design

In their statistical analysis, the authors report the relationship between the blood glucose level and age, BMI, A1C, creatinine and diabetes duration. The only variable that correlated with the blood sugar was BMI. I do not have a problem with the analysis or the presentation of the data, however, did the authors probe further to determine if a multivariate analysis would account for a greater proportion of the variance in plasma glucose? I expect that it would not, but the authors might want to add this explanation somewhere in the results section.

Validity of the findings

Having diabetes at a lower body weight can often mask other illnesses an individual possesses. I comment the authors for controlling the cause of hospital admission in the cohort, but do the authors have any indication of any differences between the low- and high-BMI? For example, some of the low BMI individuals may have undergone weight loss surgery (e.g., RYGB or sleeve-gastrectomy), which can lead to “dumping” and overt, severe hypoglycemia. Do the authors know if any of the low-BMI subjects had previously undergone hypoglycemia? If the data are available, it would be helpful to report it in the results section (i.e., the number of subjects who had previously undergone weight loss surgery). Likewise, the dose of insulin and sulfonylurea each patient was on would be informative to the reader. However, I understand if they are not available.
In the section where the authors note the limitations of the study, it should be included that they did not measure 1) liver glycogen or 2) counterregulatory hormones. The inclusion of these data in the manuscript would have markedly improved the merit of the manuscript. The reasons why you did not have these data, however, is understandable.

Additional comments

My comments are covered by the first three sections.

---

## Round 0.2 · Minor Revisions

Please address the comments suggested by reviewer 2.

Reviewer 1 ·

Basic reporting

OK

Experimental design

OK

Validity of the findings

OK

Additional comments

The authors addressed all my concerns.

Reviewer 2 ·

Basic reporting

The revisions are well done, just a couple of minor things.

Experimental design

Good.

Validity of the findings

Good.

Additional comments

In the abstract, it would be more accurate to say “This study investigates the relationship between body mass index….” Instead of “This study investigates the effect of body mass index….”

At the end of the Introduction (lines 110-112), this statement is not completely accurate. A lower BMI per se does not reflect lower liver glycogen. For example, young people with lower BMI have higher liver glycogen than older people with type 2 diabetes. I think the statement “People with T2DM who have a lower BMI are also likely to have lower hepatic glycogen stores and this can diminish the secretion of glucose counterregulatory hormones during hypoglycemia.” is more accurate.

---

## Round 0.3 · accepted · Accept

The authors have addressed all the issues raised by the reviewers.